# Laminarin Attenuates ROS-Mediated Cell Migration and Invasiveness through Mitochondrial Dysfunction in Pancreatic Cancer Cells

**DOI:** 10.3390/antiox11091714

**Published:** 2022-08-30

**Authors:** Woonghee Lee, Gwonhwa Song, Hyocheol Bae

**Affiliations:** 1Institute of Animal Molecular Biotechnology, Department of Biotechnology, College of Life Sciences and Biotechnology, Korea University, Seoul 02841, Korea; 2Department of Oriental Medicinal Biotechnology, College of Life Sciences, Kyung Hee University, Yongin 17104, Korea

**Keywords:** laminarin, pancreatic cancer, mitochondria dysfunction, migration, ROS

## Abstract

Pancreatic ductal adenocarcinoma (PDAC) is a notoriously aggressive type of cancer with a high metastasis rate. It is conventionally treated by surgical resection and neoadjuvant chemotherapy. However, continuous chemotherapy leads to relapse in most PDAC patients due to chemical resistance. Therefore, novel anticancer agents need to be identified and developed. The antitumor activities of laminarin extracted from brown algae against hepatocarcinoma, lung, and colon cancer have been established. However, its effects on pancreatic cancer have remained obscure. Our study identified the anticancer effects of laminarin on pancreatic cancer cells and tried to explain its intracellular mechanisms. We assessed the cell viability of PANC-1 and MIA PaCa-2 cells using MTT assay. Hanging drop method was used for the spheroid formation. Flow cytometry was conducted to evaluate the several intracellular alterations including apoptosis, ROS production, mitochondrial membrane potential (MMP), and calcium concentration induced by laminarin. An invasion test was performed to assess the inhibitory effect of laminarin on cell migration and the invasive genes were evaluated by RT-qPCR. Signaling pathway related with anticancer effects of laminarin was analyzed by western blot. We report that inhibiting laminarin increased the proliferation and viability of the representative pancreatic cancer cell lines, MIA PaCa-2 and PANC-1. Laminarin triggered apoptosis and mitochondrial impairment as evidenced by depolarized mitochondrial membranes, disrupted calcium, and suppressed cell migration caused by reactive oxygen species production and related intracellular signaling pathways. Moreover, laminarin showed synergistic effects when combined with 5-FU, a standard anticancer agent for PDAC. The present study is the first to report that laminarin exerts anticancer effect through ROS production in pancreatic cancer cells. Laminarin shows potential to serve as a new anticancer agent for treating PDAC.

## 1. Introduction

Pancreatic cancer (PC) is one of the most lethal cancers, ranking third in fatalities among men and women in the USA and fourth in Europe [1,2]. Pancreatic ductal adenocarcinoma (PDAC) derived from lined ductal epithelial cells is the most prevalent type of PC [3]. The median survival of PDAC is only 4.6 months from diagnosis, as it is usually identified at advanced stages owing to inadequate early detection methods and a highly metastatic and aggressive profile [4,5,6]. Predictions indicate that PDAC will become the second most prevalent cause of cancer-related mortality by 2030 [7]. Surgical resection and neoadjuvant chemotherapy are the conventional approaches for treating PDAC [8]. The standard chemotherapeutic agents for PDAC, 5-fluorouracil (5-FU) and gemcitabine, can relieve symptoms and improve the performance status of patients. However, their effects are limited by chemoresistance, which causes relapse in most patients [9,10,11]. The combination of 5-FU, leucovorin, irinotecan, and oxaliplatin (FOLFIRINOX) regimen is effective against advanced PDAC, but the adverse effects are more severe than those of gemcitabine [12]. Therefore, safer and novel anticancer drugs and effective combination therapies with mild or no complications and better effects are required.

Laminarin extracted from brown algae serves as a storage glucan for their maturation. It is a short glucan polymer that comprises (1,3)-β-D-glucan residues with a β-(1,6)-intrachain linkage and is one of the most abundant polysaccharides in the aquatic environment [13,14]. Laminarin possesses antioxidant [15], anticancer [16], and immunoregulatory [17] properties. The pharmacological properties of laminarin are more robust when chemically modified [18]. Moreover, laminarin is a dietary modulator that affects the gastrointestinal tract when ingested as a dietary fiber [19]. The role of laminarin in hepatocarcinoma, lung, and colon cancer has been actively explored [20,21,22]. However, whether laminarin has effective anticancer activity in PC has remained obscure.

The present study aimed to determine the anticancer effects of laminarin in terms of reactive oxygen species (ROS) production, apoptosis, mitochondrial impairment, calcium homeostasis, migration, and intracellular signaling pathways in the representative pancreatic cancer cell lines, MIA PaCa-2 and PANC-1.

## 2. Materials and Methods

### 2.1. Reagents and Antibodies

Laminarin isolated from *Laminaria digitata* (brown algae; Cat. No. L9634), N-acetyl-L-cysteine (NAC; Cat. No. A9165), and 5-fluorouracil (5-FU; Cat. No. F6627) (all from Sigma-Aldrich Corp., St. Louis, MO, USA) were dissolved in warm ultra-pure water and subsequently in dimethyl sulfoxide (DMSO). Appendix A shows the antibodies and their sources.

### 2.2. Cells and Culture

The human PC cell lines originating from epithelioid carcinoma of the pancreatic duct PANC-1 and MIA PaCa-2 (Korean Cell Line Bank Seoul, Korea) were cultured in Dulbecco’s Modified Eagle Medium containing 10% fetal bovine serum and 1% penicillin-streptomycin. Cultured PC cell monolayers were maintained at 37 °C under a humidified 5% CO_2_ atmosphere. Upon reaching 70% confluence, the cells were starved in serum-free medium for 24 h, then incubated with various doses of laminarin with or without NAC or 5-FU. All experiments were repeated at least three times.

### 2.3. Cell Viability

We assessed cell viability using Cell Proliferation Kit I (Cat no: 11465007001; Roche Holdings AG, Basel, Switzerland), which is a 3-(4,5-dimethylthiazol-2-yl)-2,5-diphenyl-2H-tetrazolium bromide (MTT) assay. The PC cells in 96-well culture plates were incubated with various concentrations of laminarin and 10 µL of MTT tetrazolium salt at 37 °C for 4 h, followed by incubation in solubilization buffer at 37 °C overnight. Absorbance was measured at 560 and 650 nm using a microplate reader.

### 2.4. Cell Aggregation

The hanging drop method was used to determine spheroid formation. The PC cells were maintained for 3 days with laminarin (2 mg/mL) alone or with either the ROS inhibitor NAC (0.5 mM) or the chemotherapeutic agent 5-FU (20 μM). Spheroid morphology was assessed using a DM3000 microscope (Leica Microsystems GmbH, Wetzlar, Germany). The total area and density of cell aggregation were calculated as described [23].

### 2.5. Cell Migration

We assessed the migratory capacity of PC cells in 35-mm migration culture dishes (Ibidi, Munich, Germany) as described by the manufacturer. Suspended PC cells were seeded into each well and maintained for 16 h until they reached 90% confluence. The cells were then gently rinsed with phosphate-buffered saline (PBS) and serum-starved overnight. The partition was removed from the dish surface leaving a gap of 500 µm between the cell patches. Laminarin (2 mg/mL) with or without NAC (0.5 mM) and 5-FU (20 μM) was added to the dishes, and images of the gap were acquired using the DM3000 microscope. The amount of cell migration was calculated as the gap distance.

### 2.6. Cell Invasion

Cells were seeded on SPLInsert^TM^ Hanging membranes (Cat. No. 35224; SPL Life Sciences, Pocheon, Korea); further, laminarin (2 mg/mL) with or without NAC and 5-FU was added to the wells. After 16 h, the inserts were fixed with methanol for 10 min, air-dried and stained with hematoxylin for 30 min. The membranes were washed several times with PBS, and cells on the membrane inside the insert were removed using a cotton swab. The membranes were detached from the inserts, placed on glass slides, and covered with Permount solution. Invasive cells were counted using the DM3000 microscope.

### 2.7. Mitochondrial Membrane Potential (MMP)

We assessed the loss of mitochondrial function in PC cells using a mitochondrial staining kit (Cat. no: CS0390; Sigma-Aldrich Corp.). The cells were incubated with various concentrations of laminarin with or without NAC or 5-FU. The cells were stained with JC-1 and washed with JC-1 destaining solution; further, fluorescence emission was analyzed using a flow cytometer (BD Biosciences, San Jose, CA, USA). We gated 10,000 cells in all dot plots, and the assay was repeated in triplicate.

### 2.8. Reactive Oxygen Species

We evaluated the total ROS production, including that of hydroxyl radicals (•OH) and peroxynitrite (ONOO^−^) [24], in PC cells stained with 2,7-dichlorofluorescin diacetate (DCFH-DA; Cat. No.: D6883, Sigma-Aldrich Corp.) that ROS transform into fluorescent 2,7-dichlorofluorescin (DCF). The cells were incubated with 0, 0.5, 0.8, 1, or 2 mg/mL laminarin, or 2 mg/mL laminarin plus NAC or 5-FU for 1 h and stained with DCFH-DA for 30 min at 37 °C. The supernatants were collected and harvested cells were washed with PBS. The fluorescence intensity of DCF based on 10,000 cells per gate was calculated using the flow cytometer. The assay was repeated in triplicate.

### 2.9. Mitochondrial Ca^2+^

The PC cells were seeded into 6-well plates and incubated with 0, 0.5, 0.8, 1, and 2 mg/mL of laminarin alone or with 2 mg/mL of laminarin plus NAC or 5-FU. Mitochondrial calcium (Ca^2+)^ levels in cells washed with Hank’s balanced salt solution (HBSS) were evaluated by staining with Rhod-2 AM that accumulates in the mitochondria and binds to Ca^2+^. The intensity of fluorescence emitted by Rhod-2 was assessed in 10,000 cells per gate using the flow cytometer. The assay was repeated in triplicate.

### 2.10. Intracellular Ca^2+^ Level Analysis

We seeded PC cells into 6-well plates and incubated them with 0, 0.5, 0.8, 1, and 2 mg/mL of laminarin alone or with 2 mg/mL of laminarin plus NAC or 5-FU. The cells were stained with Fluo-4 AM for 20 min, washed with PBS, and the fluorescence intensity emitted by Fluo-4 AM bound to cytosolic calcium ions was assessed in 10,000 cells per gate using the flow cytometer. The assay was repeated in triplicate.

### 2.11. Apoptosis Quantitation in PC Cells

We investigated the apoptotic effects of laminarin on PC cells using Annexin V Apoptosis Detection Kit I (BD Bioscience). The cells were incubated with 0, 0.5, 0.8, 1, and 2 mg/mL of laminarin alone or with 2 mg/mL of laminarin plus NAC or 5-FU and stained with annexin V and PI at room temperature. Fluorescence intensity was measured in 10,000 cells per gate using the flow cytometer (BD Biosciences). The assay was repeated in triplicate.

### 2.12. Western Blotting

Protein was extracted from PC cells using RIPA lysis buffer (Cat. No: R0278, Sigma-Aldrich Corp.), quantified using the Bradford reagent (Bio-Rad Laboratories Inc., Hercules, CA, USA) and resolved by SDS-PAGE. The proteins were blotted onto polyvinylidene fluoride membranes and incubated with primary antibodies at 4 °C for 16 h, followed by secondary antibodies for 1 h. Target proteins (Appendix A) were detected using West-Q Pico chemiluminescent substrate (GenDEPOT, Katy, TX, USA) and an Alliance Mini HD9 acquisition system (Alliance UVItec Ltd., Cambridge, UK).

### 2.13. Real-Time Quantitative PCR (RT-qPCR)

Total RNA was extracted from PC cells using TRIzol as described by the manufacturer using a spectrophotometer (Thermo Fisher Scientific Inc., Waltham, MA, USA). Complementary DNA was synthesized using AccuPower^®^ RT PreMix (Bioneer, Daejeon, Korea), and products of interest were amplified by RT-qPCR using SYBR green and the CFX Connect Real-Time System (Bio-Rad Laboratories Inc.) under the following temperature conditions: 95 °C for 3 min, followed by 40 cycles at 95 °C for 20 s, 64 °C for 40 s, and 72 °C for 1 min. We confirmed that only one product was amplified using a melting curve from 55 to 95 °C. Appendix A shows the specific primers designed using Primer 3 software (http://primer3.ut.ee accessed on 22 March 2022).

### 2.14. Statistical Analysis

All data were assessed by analysis of variance (ANOVA), followed by Dunnett’s post hoc test using the Statistical Analysis System (SAS, Cary, NC, USA). All experiments were performed in triplicate. Values with *p* < 0.05 were considered statistically significant. Data are presented as means ± standard deviation.

## 3. Results

### 3.1. Laminarin Inhibited Cell Growth and Triggered Apoptosis in PC Cells

The viability of PC cells was reduced with increasing concentrations of laminarin. The viability of PANC-1 and MIA PaCa-2 cells incubated with 2 mg/mL laminarin was 61% and 64%, respectively (*p* < 0.01 for both; Figure 1A,B). We previously reported that this concentration of laminarin exerted significant antiproliferative effects in ovarian cancer cells and did not cause toxicity in zebrafish embryos with respect to viability [25]. Therefore, we estimated that 2 mg/mL was the optimal dose and applied it herein. We assessed the antiproliferative effects of laminarin on PC cells in a 3D environment by analyzing spheroid formation using the hanging drop method. Laminarin (2 mg/mL) reduced the total area of spheroids formed in PANC-1 and MIA PaCa-2 cells by 73% and 62%, respectively (*p* < 0.001 for both; Figure 1C,D), and decreased the relative tumor density to 19% and 12%, respectively (*p* < 0.001 for both; Figure 1C,D).

We investigated whether laminarin exerts apoptotic effects on PC cells. The proportion of apoptotic PC cells gradually increased with increasing laminarin concentrations. At laminarin (2 mg/mL), relative late apoptosis in PANC-1 and MIA PaCa-2 cells increased to 210% (*p* < 0.01) and 185% (*p* < 0.001), respectively (Figure 1E,F).

### 3.2. Effects of Laminarin on ROS Generation and MMP in PC Cells

We previously found that laminarin triggers apoptosis in PC cells. Here we investigated the underlying mechanism by exploring whether laminarin leads to ROS generation in PC cells. Figure 2A shows that laminarin concentration-dependently increased ROS generation in PANC-1 and MIA PaCa-2 cells from 25.7% to 52.9% and from 31.5% to 59.0%, respectively (*p* < 0.01 for both vs. positive control with hydrogen peroxide). We then analyzed mitochondrial function by measuring MMP loss. Figure 2B shows increased depolarization of the mitochondrial membrane in response to laminarin. The proportion of JC-1 monomers significantly increased to 257% and 276% in PANC-1 and MIA PaCa-2 cells (*p* < 0.001 for both vs. positive control). Collectively, laminarin elevated ROS production and decreased MMP levels in PC cells.

### 3.3. Effects of Laminarin on Calcium Ion Flow between Mitochondria and Cytoplasm in PC Cells

We verified the effect of laminarin on calcium ion regulation in PC by quantifying mitochondrial and cytosolic calcium levels, respectively, using Rhod-2 and Fluo-4 assays. Flow cytometry data showed that decreased effect of laminarin (2 mg/mL) on mitochondrial calcium level was 50% and 52% in PANC-1 and MIA PaCa-2 cells, respectively (*p* < 0.001 for both; Figure 2C) and significantly increased cytosolic calcium levels to 183% and 186%, respectively (*p* < 0.001 for both; Figure 2D). These data suggested that laminarin can cause cytosolic calcium overload in PC cells.

### 3.4. Signal Transduction Associated with Anticarcinogen Effects of Laminarin in PC Cells

We investigated signaling pathways in PC cells involved in the anti-cancer effects of laminarin by quantifying mitogen-activated protein kinase (MAPK) and Ak strain transforming (AKT) phosphorylation and Kirsten rat sarcoma viral oncogene homolog (KRAS) activity. Laminarin (2 mg/mL) increased the abundance of phosphorylated AKT and JNK in PANC-1 cells ~3- and >3.5-fold, respectively (*p* < 0.001 for both vs. control; Figure 3A) and in MIA PaCa-2 cells by 1.8- and ~1.5-fold (*p* < 0.001 and *p* < 0.01, respectively; Figure 3B). In contrast, laminarin (2 mg/mL) inhibited ERK1/2 phosphorylation to 0.1- and 0.2-fold in PANC-1 and MIA PaCa-2 cells (*p* < 0.001 and *p* < 0.01 vs. control, respectively; Figure 3C), and phosphorylated P38 was 0.13- and 0.27-fold in PANC-1 and MIA PaCa-2 cells, respectively (*p* < 0.001 for both vs. control; Figure 3D). Meanwhile, laminarin minimally affected KRAS activity in both cell lines (Figure 3E). These results imply that laminarin regulates MAPK and AKT signaling pathway.

### 3.5. Laminarin Attenuated PC Cell Invasiveness

Transwell invasion and migration assays revealed that laminarin inhibited cell invasion and migration, the two mechanisms responsible for PC metastasis and cell growth. The results of the Transwell^TM^ cell invasion assays showed that suppressive effect of laminarin (2 mg/mL) in cell invasion was 16% and 21% in PANC-1 and MIA PaCa-2 cells, respectively (*p* < 0.01 for both vs. untreated controls; Figure 4A,B). Consistent with these results, laminarin also inhibited PC cell migration (Figure 4C) and increased the interspace gap between the cell populations up to ~150% (*p* < 0.001) and ~127% (*p* < 0.01) in PANC-1 and MIA PaCa-2 cells, respectively, compared with the control (Figure 4D). We then evaluated the transcription of invasive genes by qRT-PCR (Figure 4E,F). Laminarin (2 mg/mL) significantly and slightly, but non-significantly reduced the expression of the *forkhead box protein M1* (*FOXM1*) and *vascular endothelial growth factor A* (*VEGFA*) genes in PANC-1 and MIA PaCa-2 cells, respectively. Laminarin (2 mg/mL) significantly increased the expression of *cadherin-1* (*CDH1*) and *tissue inhibitor of metallopeptidase1* (*TIMP1*) genes. These findings show that laminarin attenuated PC cell invasiveness.

### 3.6. Laminarin-Induced ROS Production Regulated PC Cell Proliferation

Because ROS are important intracellular signaling molecules that regulate the physiological and pathological progression of various cells [26], we assessed PC cell viability in response to laminarin using the ROS scavenger NAC. N-acetyl-L-cysteine significantly increased the relative viability of cells incubated with laminarin from 64% to 76% (*p* < 0.001) and from 63% to 88% (*p* < 0.05) in PANC-1 and MIA PaCa-2 cells, respectively (Figure 5A,B). Spheroids formed in PC cells incubated with laminarin and NAC (Figure 5C,E). The decreased total area and density of spheroids induced by laminarin were considerably recovered by NAC (Figure 5D,F). These results suggest that laminarin-induced ROS production regulates PC cell progression.

### 3.7. Laminarin-Mediated ROS Production Led to Apoptotic Cell Death and Regulated Mitochondrial Function and Calcium Homeostasis in PC Cells

As it was observed that ROS production induced by laminarin influenced PC cell proliferation, we postulated that laminarin-mediated ROS production causes apoptosis. Laminarin-induced ROS were attenuated by NAC in PC cells (Figure 6A), whereas NAC significantly attenuated the increase in late apoptotic cells induced by laminarin from 248% to 96% and from 229% to 128% in PANC-1 and MIA PaCa-2 cells, respectively (*p* < 0.01 for both; Figure 6B).

We explored the effect of laminarin-mediated ROS production on mitochondrial function by analyzing the loss of MMP in the presence of NAC. Mitochondrial membrane depolarization increased by laminarin was recovered by NAC from 166% to 113% (*p* < 0.05) and 224% to 140% (*p* < 0.01) in PANC-1 and MIA PaCa-2 cells, respectively (Figure 6C). We then assessed whether ROS production induced by laminarin could influence mitochondrial and cytoplasmic calcium ion levels in PC cells incubated with laminarin and NAC using Rhod-2 and Fluo-4 assays, respectively. The reduced mitochondrial calcium levels induced by laminarin were significantly increased by NAC from 63% to 90% (*p* < 0.01) and from 57% to 78% (*p* < 0.001) in PANC-1 and MIA PaCa-2 cells, respectively (Figure 6D). In contrast, NAC attenuated cytosolic calcium overload induced by laminarin from 177% to 130% (*p* < 0.001) and from 166% to 138% in PANC-1 and MIA PaCa-2 cells, respectively (Figure 6E). Collectively, laminarin-mediated ROS production triggered apoptosis and regulated MMP and calcium homeostasis in PC cells.

### 3.8. Laminarin Regulated ROS-Induced Signaling Transduction and PC Cell Migration

We applied western blotting to examine the relationship between ROS production and MAPK and AKT signaling pathways in cells incubated with NAC followed by laminarin. The increased phosphorylation of AKT and JNK was inhibited in both the cell lines (Figure 7A,B). The phosphorylation of ERK decreased by laminarin was restored by NAC in PANC-1 cells, but its effects were limited in MIA PaCa-2 cells (Figure 7C). The phosphorylation of P38 diminished by laminarin was significantly improved by NAC (Figure 7D). We then assessed associations between cell invasiveness and ROS production in PC cells using migration and Transwell^TM^ (SPL Life Sciences, Pocheon, Korea) invasion assays. The number of PC cells that passed through the membrane in the Transwell^TM^ assay was slightly increased by NAC followed by laminarin compared with laminarin alone, but with no statistical significance (Figure 7E). In the migration assay, the interspace gap between divided PANC-1 cell populations increased by laminarin was significantly diminished by NAC from 220% to 200% (*p* < 0.05; Figure 7F). These findings indicate that laminarin regulates ROS signal transduction and ROS-mediated cell migration in PC cells.

### 3.9. Effects of Laminarin and Standard Anticancer Drug Were Synergistic in PC Cells

We assessed the antitumor effects of laminarin combined with the standard anticancer agent, 5-FU. Figure 5 shows that laminarin significantly attenuated cell viability and inhibited the formation of PC cell spheroids when combined with 5-FU. The combination produced significant amounts of ROS (Figure 6A) that led to apoptosis (Figure 6B), decreased MMP (Figure 6C), and disrupted calcium homeostasis (Figure 6D,E) compared with 5-FU alone. Moreover, the phosphorylation levels of AKT and JNK were bolstered and those of ERK1/2 and p-P38 were conversely weakened by the combination of 5-FU and laminarin vs. 5-FU alone (Figure 7A–D). Furthermore, the inhibitory effect on cell invasion and migration was strengthened by the combination of 5-FU and laminarin vs. 5-FU alone (Figure 7E,F).

## 4. Discussion

Our findings indicate that laminarin exerts anticancer activity in PC cells. The intracellular mechanisms associated with the anticancer activity of laminarin and its synergistic effects with conventional chemotherapeutic agents in PC have not been clearly explained until now. We verified that laminarin inhibits the progression and proliferation of PC cells and induces ROS-mediated apoptosis and mitochondrial dysfunction. Moreover, laminarin-induced ROS production regulated signaling pathways associated with anticancer effects as well as mitochondrial and cytoplasmic calcium ion concentrations. Furthermore, laminarin attenuated cell migration and invasion by regulating genes associated with invasion and exerted synergistic effects when combined with the standard anticancer adjuvant 5-FU. Therefore, our findings suggested that laminarin shows potential as a novel chemotherapeutic agent against PC.

Although laminarin is an exogenous polysaccharide, previous reports have suggested that it exerts anticancer effects in various types of cancers by activating apoptosis. For example, laminarin suppresses proliferation and promotes apoptosis in Bel-7404 and HepG2 cells and in hepatocellular carcinoma [20]. The inhibition of lung cancer cell proliferation and migration confirmed its antitumor activity [21]. Apoptotic effects of laminarin in LOVO and HT-29 cells as well as in human colon cancer cells have been confirmed by the expression of proteins associated with apoptosis and signaling pathways [22,27]. We previously showed that laminarin also regulates cell progression via apoptosis induced by mitochondrial dysfunction and endoplasmic reticulum (ER) stress in ovarian cancer cells [25]. The present study confirmed that the suppression of cell proliferation and migration, the activation of apoptosis, and signaling transduction are associated with anticancer effects and mitochondrial dysfunction. These findings suggest that laminarin triggers apoptosis in PC by influencing diverse intracellular modes of action.

The aggressive profile of PDAC is attributed to its ability to metastasize to adjacent organs, including the liver and gallbladder [28]. Laminarin significantly reduced the metastatic potential of PC cells as per the Transwell^TM^ cell invasion and migration assays. The epithelial-mesenchymal transition (EMT) is considered the initial step for metastatic dissemination. Cell dissociation from the epithelial surface leads to dysregulated cell-cell communication and enhanced invasive ability [29]. Many factors that are closely associated with the EMT in PDAC such as *FOXM1*, *VEGFA*, *CDH1*, and *TIMP1*, are interconnected and involved in metastasis and progression [30,31]. Forkhead box protein M1 is a critical factor for PDAC to acquire the EMT phenotype [32]. The expression of *VEGFA* is involved in tumor proliferation and motility in PDAC, and a higher abundance is associated with a worse prognosis [33,34]. Moreover, the expression of E-cadherin encoded by *CDH1*, a marker of the epithelial cell phenotype, plays an important role in suppressing cell invasion [35]. The depletion of E-cadherin is typical of the EMT and it can trigger PDAC cell migration [36]. As a key gene involved in the EMT, *TIMP1* participates in repressing PANC-1 cell migration [37]. The present qRT-PCR findings confirmed expression of the invasive genes *FOXM1*, *VEGFA*, *CDH1*, and *TIMP1*. Overall, the above results show that laminarin suppressed cell migration through the regulation of invasive genes in PC cells.

Reactive oxygen species play important roles in cancer progression and growth. Compared with normal cells, many types of cancer cells have increased basal levels of ROS owing to disrupted redox homeostasis induced by oxidative stress [38]. However, excessive ROS production is as toxic to cancer as it is to normal counterpart cells and this can be enhanced using exogenous agents that induce further ROS production [39]. Therefore, regulating redox status has been considered an effective strategy for eradicating cancer cells [38]. The elimination of PDAC via ROS regulation has been addressed in various ways. Synthetic triterpenoids derived from oleanolic acid exert ROS-dependent antiproliferative and apoptotic effects on PC cells [40]. Some therapeutic agents such as resveratrol, spiclomazine, and SKLB316 decrease MMP through ROS accumulation in PC [41,42,43]. Moreover, longikaurin E triggers PC cell apoptosis by regulating P38 and PI3K/AKT pathways driven by ROS. The present findings are in line with the fact that the ROS scavenger NAC reverses these effects [44]. Furthermore, mahanine elevates the intracellular calcium concentrations induced by ROS, which leads to ER stress in pancreatic adenocarcinoma cells, and pretreatment with NAC diminishes such elevation [45]. Upregulated ROS can be involved in PC tumor progression and migration. For example, ROS-driven programmed cell death attenuates PC migration independently of the Hippo pathway [46]. Anthocyanins also inhibit PC cell migration through ROS production [47]. The present findings also suggest that ROS inhibit cell proliferation, drive mitochondrial dysfunction, regulate MAPK and AKT signaling pathways, disrupt calcium homeostasis, and suppress PC cell migration. The effects induced by laminarin were reversed by treatment with NAC. Therefore, the anticancer effects of laminarin are accomplished via ROS production.

Permanent activation of the *KRAS* oncogene is closely involved in the maintenance of cell proliferation, migration, transformation, and survival of PDAC [48]. Efforts to directly apply point mutations of *KRAS* as clinical therapy have failed [49]. Currently, the rapidly accelerated fibrosarcoma-mitogen activated protein kinase kinase-extracellular signal-regulated kinase (RAF-MEK-ERK) pathway and molecules downstream of KRAS or PI3K-AKT-mammalian target of rapamycin (mTOR) signaling are targeted to eliminate PDAC. Inhibitors of the RAF-MEK-ERK pathway exert synergistic anticarcinogenic effects with a PI3K pathway inhibitor in mouse models [48]. Here, we investigated whether laminarin attenuated activated *KRAS* in PDAC. We found that laminarin influenced the MAPK signaling pathway and AKT but not KRAS protein levels. That is, laminarin increased the phosphorylation of JNK and AKT and inhibited that of ERK1/2 and P38. These results suggested that the anticancer effects of laminarin are *KRAS*-independent.

Colon, breast, and pancreatic cancers are generally treated with 5-FU [50,51]. Although 5-FU is potent against several types of cancer, chemoresistance and adverse effects have become a concern [9]. The present findings of cell viability, MMP, apoptosis, mitochondrial and cytoplasmic calcium content, and cell migration support the hypothesis that laminarin exerts synergistic ability with 5-FU in PC. Therefore, we concluded that the effects of laminarin are synergistic when combined with conventional anticancer drugs for PDAC. Finally, these findings can support further studies in vivo, with respect to determining the optimal doses of laminarin and evaluating synergistic effect with anticancer agents to treat PDAC. However, we were unable to identify which factors correlated with the synergistic effects. Further studies are required to determine the associated proteins and genes.

## 5. Conclusions

The present study revealed the underlying mechanisms through which laminarin causes apoptotic cell death and suppresses PC cell migration. We confirmed that laminarin-inhibited cell proliferation and migration, triggered mitochondrial dysfunction, and disrupted calcium homeostasis via ROS production. Furthermore, these effects were enhanced when laminarin was combined with 5-FU. Therefore, laminarin could be a potential therapeutic agent and supplementary chemotherapy for PDAC, considering these synergetic effects. Although our findings in vitro are limited, they might serve as the basis for further studies in vivo and the development of an innovative agent for treating PDAC.

## Figures and Tables

**Figure 1 antioxidants-11-01714-f001:**
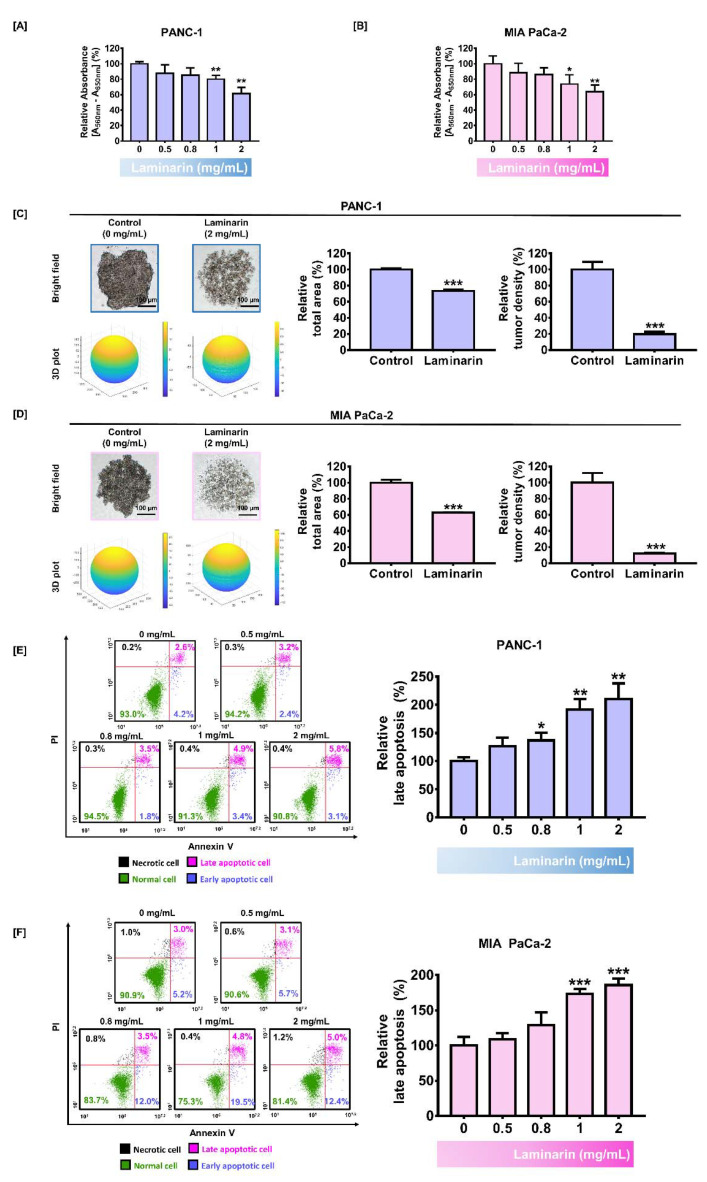
Antitumor effects of laminarin on PC cells. (**A**,**B**) Relative cell proliferation decreases with increasing laminarin concentrations of 0, 0.5, 0.8, 1, and 2 mg/mL. (**C**,**D**) Spheroid formation in cells without and with laminarin. Scale bar: 100 µm. (**E**,**F**) Apoptotic cell death analyzed using Annexin V and propidium iodide. Purple dots (upper right) in dot plots indicated late apoptotic cells. (* *p* < 0.05, ** *p* < 0.01, and *** *p* < 0.001; laminin vs. control). All experiments were conducted in triplicate.

**Figure 2 antioxidants-11-01714-f002:**
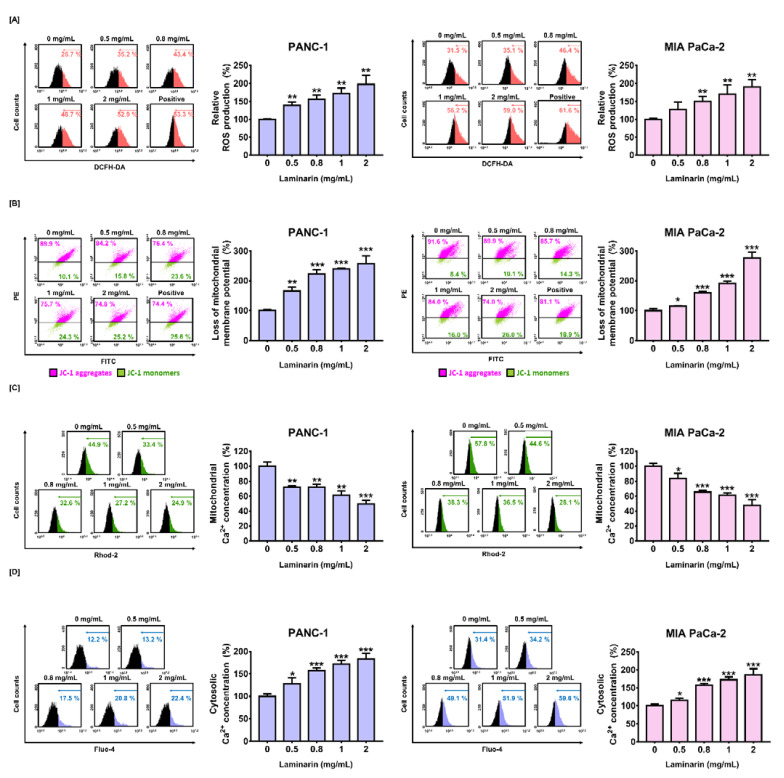
Dose-dependent anticancer effects of laminarin on ROS production, MMP, and calcium homeostasis in PC cells. (**A**) Intracellular ROS evaluated as DCF fluorescence emission by flow cytometry with H_2_O_2_ (100 µM) as the positive control. (**B**) JC-1 monomers in PC cells detected by flow cytometry. Purple and green dots in plot represent JC-1 aggregates and monomers, respectively. Valinomycin (1 µg/mL) was the positive control. (**C**) Calcium ion concentrations in mitochondria analyzed using Rhod-2. (**D**) Changes in cytosolic calcium concentrations induced by laminarin detected with Fluo-4. (* *p* < 0.05, ** *p* < 0.01, *** *p* < 0.001; laminin vs. control). All experiments were conducted in triplicate.

**Figure 3 antioxidants-11-01714-f003:**
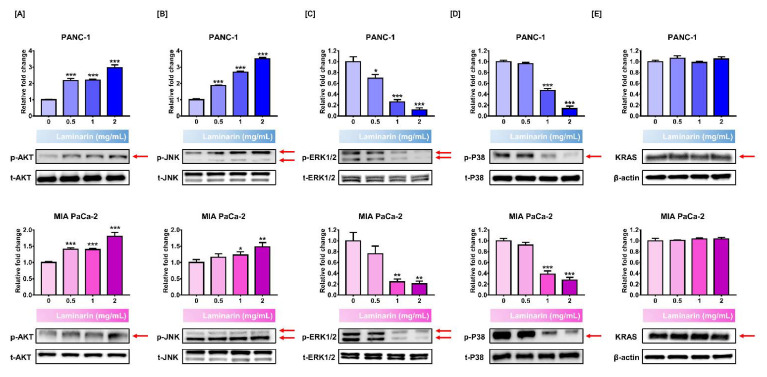
Effect of laminarin on signaling pathway associated with antitumor ability in PC cells. Protein levels of (**A**) p-AKT, (**B**) p-JNK, (**C**) p-ERK1/2, and (**D**) p-P38 (**E**) and KRAS expression. (* *p* < 0.05, ** *p* < 0.01, *** *p* < 0.001; laminin vs. control). Red arrows indicate each band of protein which is quantified. All experiments were conducted in triplicate. PC, pancreatic cancer.

**Figure 4 antioxidants-11-01714-f004:**
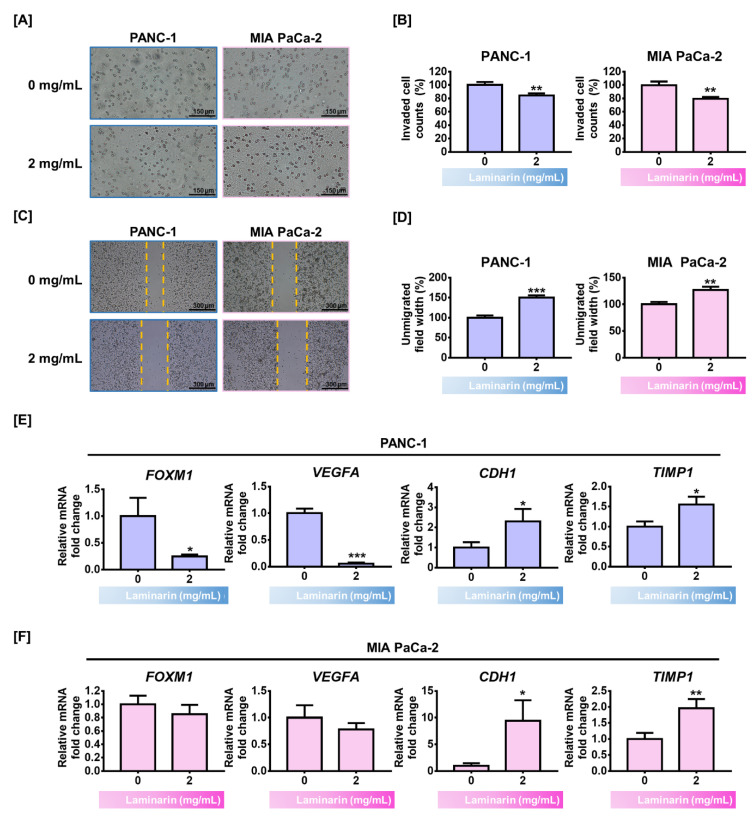
Inhibitory effects of laminarin on cell migration and invasion in PC cells. (**A**,**B**) Transwell invasion assays of PC cell invasion. Graph shows relative numbers of cells that passed through membrane. Scale bar: 150 µm. (**C**,**D**) Cell migration in Ibidi 35 mm culture dishes. Graph shows relative gap between divided cell populations. Scale bar: 300 µm. (**E**,**F**) Invasive *FOXM1*, *VEGFA*, *CDH1*, and *TIMP1* gene expression in PC cells determined by qRT-PCR. (* *p* < 0.05, ** *p* < 0.01, *** *p* < 0.001; laminin vs. control). All experiments were conducted in triplicate.

**Figure 5 antioxidants-11-01714-f005:**
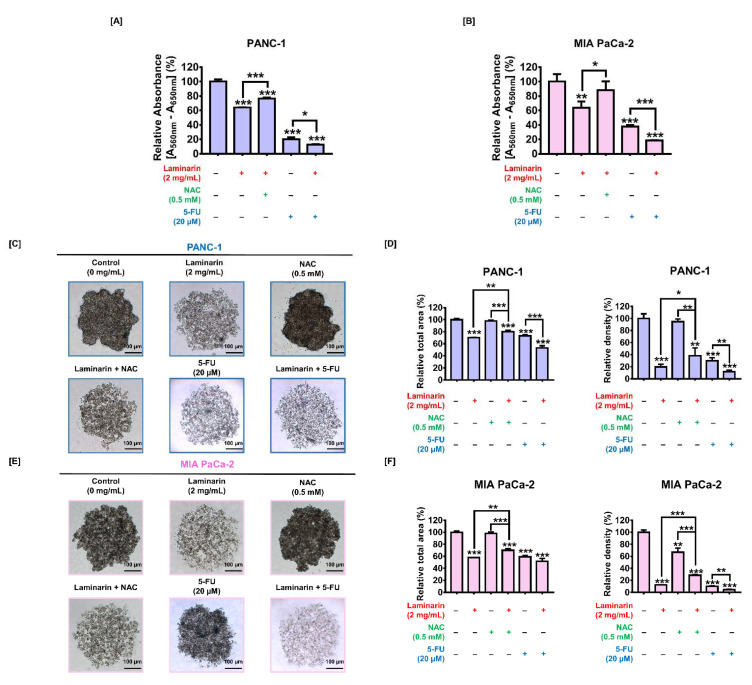
Effects of laminarin-induced ROS production and of laminarin combined with 5-FU on cell proliferation. Viability (**A**,**B**) and spheroid formation (**C**–**F**) of PC cells incubated with laminarin alone or with NAC or 5-FU. Relative spheroid density was calculated using ImageJ software. (* *p* < 0.05, ** *p* < 0.01, and *** *p* < 0.001). Scale bar: 100 µm. All experiments were conducted in triplicate. 5-FU, 5-fluorouracil; NAC, N-acetyl-L-lysine; PC, pancreatic cancer; ROS, reactive oxygen species.

**Figure 6 antioxidants-11-01714-f006:**
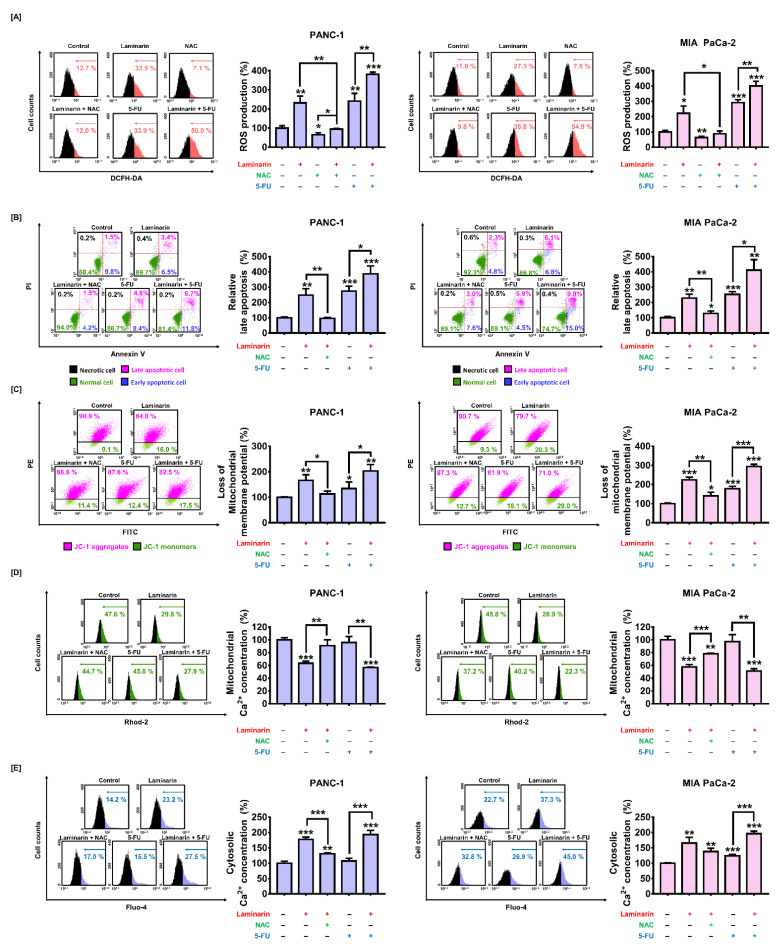
Synergistic effects of laminarin (2 mg/mL) and 5-FU (20 µM) in PC cells. Effects on (**A**) ROS production, (**B**) apoptosis, (**C**) MMP, (**D**) mitochondrial, and (**E**) cytosolic calcium concentrations. (* *p* < 0.05, ** *p* < 0.01, and *** *p* < 0.001). All experiments were conducted in triplicate. 5-FU, 5-fluorouracil; MMP, mitochondrial membrane potential; PC, pancreatic cancer; ROS, reactive oxygen species.

**Figure 7 antioxidants-11-01714-f007:**
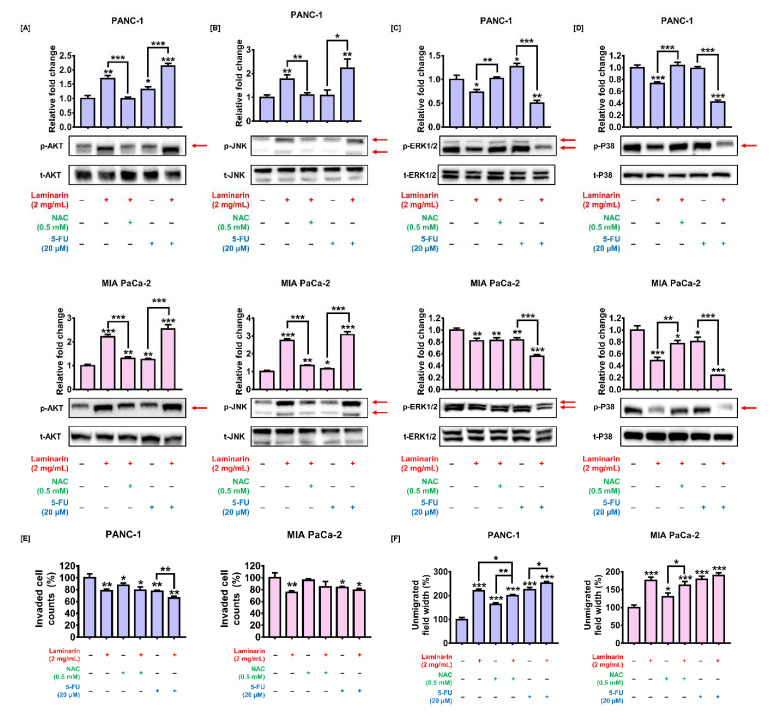
Synergistic ability of laminarin with 5-FU and effects of laminarin-induced ROS on cell invasion. (**A**–**D**) Effects of laminarin (2 mg/mL) alone or with NAC (0.5 mM) or 5-FU (20 µM) on phosphorylation (**A**) AKT, (**B**) JNK, (**C**) ERK1/2, and (**D**) P38. Red arrows indicate each band of protein which is quantified. (**E**,**F**) Cell invasion and migration after incubation with laminarin alone or with NAC or 5-FU. (* *p* < 0.05, ** *p* < 0.01, and *** *p* < 0.001). All experiments were conducted in triplicate. 5-FU, 5-fluorouracil; NAC, N-acetyl-L-lysine; PC, pancreatic cancer; ROS, reactive oxygen species.

## Data Availability

Not applicable.

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
