# Peer review of "Laminarin Attenuates ROS-Mediated Cell Migration and Invasiveness through Mitochondrial Dysfunction in Pancreatic Cancer Cells"

_antioxidants, 2022, doi:10.3390/antiox11091714_

Round 1
Reviewer 1 Report
In the present study Lee et al showed the therapeutic potential of Laminarin in PDAC. The authors provided convincing data, however there some issues which should be addressed.
1. The tables for antibody and primers should be moved to a supplementary file.
2. Scale bars are missing in Fig.1C, 1D, 5C, 5E. They should be included.
3. For western blots where there are multiple bands for a given protein it's difficult to understand the data. Therefore, the authors should clearly mark with arrows which bands correspond to each of the protein that are quantified. Also, for the raw blot images in supplementary figures, each protein should be marked by the arrows clearly indicating the band.
4. For all the experiments the authors should clearly mention how many times each experiment was done.
5. In the abstract the authors wrote that- The graphical abstract shows... I don't see the graphical abstract in the files submitted.
Author Response
[Comments and Suggestions for Authors]
[Reviewer 1]
In the present study Lee et al showed the therapeutic potential of Laminarin in PDAC. The authors provided convincing data, however there some issues which should be addressed.
Response: We appreciate the reviewer’s valuable comments and suggestions on our manuscript. We have substantially revised our manuscript according to the reviewer’s suggestions. To address the reviewer’s comments, we prepared a point-by-point response to each comment of the reviewer and highlighted changes in text of the manuscript in yellow.
Comments
- The tables for antibody and primers should be moved to a supplementary file
Response: We appreciate the reviewer’s valuable comments. According to reviewer’s suggestion, we moved all tables for antibody and primers to a supplementary file. Also, we revised our manuscript in line 71 and 163.
- Scale bars are missing in Fig.1C, 1D, 5C, 5E. They should be included
Response: We appreciate the reviewer’s valuable comments. According to reviewer’s suggestion, we added scale bars in each figure and figure legends.
- For western blots where there are multiple bands for a given protein it's difficult to understand the data. Therefore, the authors should clearly mark with arrows which bands correspond to each of the protein that are quantified. Also, for the raw blot images in supplementary figures, each protein should be marked by the arrows clearly indicating the band
Response: We appreciate the reviewer’s valuable comments. The phosphorylation levels of each protein (AKT, JNK, ERK1/2, and P38) were quantified by using each total protein (t-AKT, t-JNK, t-ERK1/2, and t-P38). The protein expression of KRAS was normalized by the expression of β-actin. Moreover, the phosphorylation levels of protein showing double bands in western blots (for example, p-JNK, and p-ERK1/2) were calculated by adding together the intensity of both bands. According to reviewer’s suggestion, we added red arrows in each band of protein which is quantified and the information in line 240 and 330 of figure legends.
- For all the experiments the authors should clearly mention how many times each experiment was done
Response: We appreciate the reviewer’s valuable comments. According to reviewer’s suggestion, we clearly mentioned all experiments were conducted in triplicate in all figure legends.
- In the abstract the authors wrote that- The graphical abstract shows... I don't see the graphical abstract in the files submitted.
Response: We appreciate the reviewer’s valuable comments. According to reviewer’s suggestion, we revised that sentence in our manuscript.
Reviewer 2 Report
The manuscript by Lee et al., shows that Laminarin, a polysaccharide found in brown algae possess anticancer activity in pancreatic cell lines. Their in vitro experiments reveal that Laminarin anticancer activity is mediated by its ability to induce mitochondrial dysfunction and ROS production. Please see minor comments noted below.
Comments
1. Laminarin was shown to be effective in majority of the assays performed in this study even at concentration of 0.5 mg/mL yet 2 mg/mL was used in some experiments justified based on a previous study, please comment?
2. Inconsistent usage of cell lines. In some places cell lines were referred to as PC cells, other places referred to as PANC-1 and MIA PaCa-2 cells. Please try to be consistent to throughout the manuscript.
3. In the methods: Western blotting please state the targets assessed or refer to the tables included in the methods rather than simply refer to them as ‘target proteins’
4. Statistical analysis: Please include the post hoc tested used.
5. Figures: Control groups were referred to as ‘naïve’ in the data graphs, ‘control’ in the legends. Please revise to make it consistent.
6. Line 121: Do authors mean ROS species: Peroxynitrate (ONOO-). Please revise.
7. Line 202: What was the concentration of hydrogen peroxide used (positive control for ROS & membrane potential)
8. Please elaborate on what do authors mean by laminarin causes calcium overload in cytosol. Increased mitochondrial ROS, reduced mitochondrial function often associated with increased mitochondrial calcium overload. Please elaborate on Laminarin effects on cytosolic vs mitochondrial calcium.
9. Line 236: please elaborate on ‘PC cell progression’
10. Line 245: The statement ‘TranswellTM cell invasion assays showed that laminarin (2 mg/mL) significantly suppressed cell invasion by ~84% and 79%’ sounds misleading. As the control groups were shown at 100%, the actual effects seen 16% and 21% respectively. Similar description is seen in the other sections of results. Please revise.
11. Please consider choosing better color combinations in the data graphs. Some colors make it hard to read text (i.e., yellow)
12. Line 283: typo error ‘We As’
13. Line 341: No graphical abstract was provided with the manuscript.
14. Line 342: Please revise ‘laminarin exerts anticancer activity against PC cells’ to ‘laminarin exerts anticancer activity in PC cells’
15. Authors show that Laminarin inhibits cancer cell proliferation and invasiveness via mitochondrial dysfunction and activation of apoptosis, they should consider adding a schematic depicting the events mediated by Laminarin and how these are specific to cancer cells opposed to normal cells.
16. Please add to the discussion how these results support assessing Laminarin anticancer effects in in vivo models of pancreatic cancer.
17. Experimental n values are missing. Please indicate n values in the figure legends and methods.
Author Response
[Comments and Suggestions for Authors]
[Reviewer 2]
The manuscript by Lee et al., shows that Laminarin, a polysaccharide found in brown algae possess anticancer activity in pancreatic cell lines. Their in vitro experiments reveal that Laminarin anticancer activity is mediated by its ability to induce mitochondrial dysfunction and ROS production. Please see minor comments noted below.
Response: We appreciate the reviewer’s valuable comments and suggestions on our manuscript. We have substantially revised our manuscript according to the reviewer’s suggestions. To address the reviewer’s comments, we prepared a point-by-point response to each comment of the reviewer and highlighted changes in text of the manuscript in yellow.
Comments
- Laminarin was shown to be effective in majority of the assays performed in this study even at concentration of 0.5 mg/mL yet 2 mg/mL was used in some experiments justified based on a previous study, please comment
Response: We appreciate the reviewer’s valuable comments. As the reviewer mentioned, laminarin was shown to be effective in most of the assays even at the concentration of 0.5 mg/mL. We determined 2 mg/mL of laminarin as an optimal dose for the present study, based on the cell viability test using MTT assay (Figure 1A, 1B). The 0.5 mg/mL of laminarin did not exhibit anti-proliferative and apoptotic effect in both PC cells. In addition, in our previous study, 2 mg/mL of laminarin exhibited significant anticancer effect in ovarian cancer cell lines and did not show toxicity in zebra fish embryo [1]. Moreover, 1.6 mg/mL of laminarin exhibited apoptotic effect in human colon cancer cells [2]. Furthermore, Weihua Jin et al has reported that the IC50 values of laminarin were approximately 2.7 mg/mL in human lung cancer cell line, A549 [3]. Therefore, 2 mg/mL of laminarin, the optimal dose we set in the present study, can be justified.
- Inconsistent usage of cell lines. In some places cell lines were referred to as PC cells, other places referred to as PANC-1 and MIA PaCa-2 cells. Please try to be consistent to throughout the manuscript
Response: We appreciate the reviewer’s valuable comments. According to reviewer’s suggestion, we tried to change ‘PANC-1 and MIA PaCa-2’ to ‘PC’ for consistent usage of cell lines, except for in the case of showing data values.
- In the methods: Western blotting please state the targets assessed or refer to the tables included in the methods rather than simply refer to them as ‘target proteins
Response: We appreciate the reviewer’s valuable comments. According to reviewer’s suggestion, we added the target proteins in section 2.12. linked to the Table S1 in supplementary materials.
- Statistical analysis: Please include the post hoc tested used
Response: We appreciate the reviewer’s valuable comments. According to reviewer’s suggestion, we added the information for the post hoc test in section 2.14.
- Figures: Control groups were referred to as ‘naïve’ in the data graphs, ‘control’ in the legends. Please revise to make it consistent
Response: We appreciate and agree with the reviewer’s valuable comments. According to reviewer’s suggestion, we revised ‘naïve’ in all figures to ‘control’ to make it consistent.
- Line 121: Do authors mean ROS species: Peroxynitrate (ONOO-). Please revise
Response: We appreciate and agree with the reviewer’s valuable comments. According to reviewer’s suggestion, we revised it to peroxynitrite with a reference [4].
- Line 202: What was the concentration of hydrogen peroxide used (positive control for ROS & membrane potential
Response: We appreciate and agree with the reviewer’s valuable comments. According to reviewer’s suggestion, we added the concentration of hydrogen peroxide and valinomycin in line 217 and 219, respectively.
- Please elaborate on what do authors mean by laminarin causes calcium overload in cytosol. Increased mitochondrial ROS, reduced mitochondrial function often associated with increased mitochondrial calcium overload. Please elaborate on Laminarin effects on cytosolic vs mitochondrial calcium.
Response: We appreciate the reviewer’s valuable comments. As the reviewer’s mentioned, it has been well known that increased mitochondrial ROS and mitochondrial dysfunction is associated with mitochondrial calcium overload. However, a contradictory report has recently highlighted the complex role of calcium ion. Inhibition of calcium ion transfer from endoplasmic reticulum to mitochondria can impair cancer growth [5]. Moreover, various cytotoxic agents modulated calcium homeostasis and induced calcium overload in cytosol [6]. For example, cisplatin, one of the most commonly used as chemotherapeutic agents, triggers increased cytosolic calcium levels and apoptosis [7]. Therefore, laminarin can exert anti-cancer effect through regulating calcium ion in cytosol and mitochondria.
- Line 236: please elaborate on ‘PC cell progression’
Response: We appreciate the reviewer’s valuable comments. It has been reported that MAPK and PI3K signaling pathways are deeply involved in the development and progression of pancreatic ductal adenocarcinoma (PDAC) [8, 9]. Therefore, we investigated the MAPK and AKT signaling pathways induced by laminarin in PC cells. According to reviewer’s suggestion, we revised our manuscript in line 234-235 more exactly.
- Line 245: The statement ‘TranswellTM cell invasion assays showed that laminarin (2 mg/mL) significantly suppressed cell invasion by ~84% and 79%’ sounds misleading. As the control groups were shown at 100%, the actual effects seen 16% and 21% respectively. Similar description is seen in the other sections of results. Please revise
Response: We appreciate and totally agree with the reviewer’s valuable comments. According to reviewer’s suggestion, we revised a vague description in line 208 and 245.
- Please consider choosing better color combinations in the data graphs. Some colors make it hard to read text (i.e., yellow
Response: We appreciate and totally agree with the reviewer’s valuable comments. According to reviewer’s suggestion, we changed the vague colors to more vivid colors in the data graphs.
- Line 283: typo error ‘We As
Response: We appreciate the reviewer’s valuable comments. According to reviewer’s suggestion, we revised ‘We As’ to ‘As’ in line 285 of our manuscript.
- Line 341: No graphical abstract was provided with the manuscript
Response: We appreciate the reviewer’s valuable comments and are very sorry that we forgot the graphical abstract by mistake. According to reviewer’s suggestion, we will submit the graphical abstract with our revised manuscript.
- Line 342: Please revise ‘laminarin exerts anticancer activity against PC cells’ to ‘laminarin exerts anticancer activity in PC cells
Response: We appreciate the reviewer’s valuable comments. According to reviewer’s suggestion, we revised our manuscript in line 347.
- Authors show that Laminarin inhibits cancer cell proliferation and invasiveness via mitochondrial dysfunction and activation of apoptosis, they should consider adding a schematic depicting the events mediated by Laminarin and how these are specific to cancer cells opposed to normal cells
Response: We appreciate the reviewer’s valuable comments and totally agree with the reviewer’s suggestion. First, in our previous study, 2 mg/mL of laminarin did not exhibit toxicity in zebrafish embryo in vivo [1]. Moreover, laminarin treatment did not affect the cell viability of rat small intestine epithelial cells (IEC-6) even at the concentration of 5 mg/mL [10]. Despite these several references, additional experiments with normal pancreatic cells should be conducted as reviewer mentioned. Therefore, in a follow-up study, we will make sure to add an experiment with normal pancreatic cells to argue our hypothesis more convincingly.
- Please add to the discussion how these results support assessing Laminarin anticancer effects in in vivo models of pancreatic cancer.
Response: We appreciate the reviewer’s valuable comments. According to reviewer’s suggestion, we added how our findings can support the further studies in vivo in line 429-431 of discussion part.
- Experimental n values are missing. Please indicate n values in the figure legends and methods
Response: We appreciate the reviewer’s valuable comments. According to reviewer’s suggestion, we clearly mentioned all experiments were conducted in triplicate in all figure legends and methods 2.14. section.
- Bae, H., et al., Laminarin-Derived from Brown Algae Suppresses the Growth of Ovarian Cancer Cells via Mitochondrial Dysfunction and ER Stress. Mar Drugs, 2020. 18(3).
- Ji, C.F. and Y.B. Ji, Laminarin-induced apoptosis in human colon cancer LoVo cells. Oncol Lett, 2014. 7(5): p. 1728-1732.
- Jin, W., et al., Structural analysis of a glucoglucuronan derived from laminarin and the mechanisms of its anti-lung cancer activity. Int J Biol Macromol, 2020. 163: p. 776-787.
- Kalyanaraman, B., et al., Measuring reactive oxygen and nitrogen species with fluorescent probes: challenges and limitations. Free Radic Biol Med, 2012. 52(1): p. 1-6.
- Cardenas, C., et al., Selective Vulnerability of Cancer Cells by Inhibition of Ca2+ Transfer from Endoplasmic Reticulum to Mitochondria. Cell Reports, 2016. 14(10): p. 2313-2324.
- Danese, A., et al., Calcium regulates cell death in cancer: Roles of the mitochondria and mitochondria-associated membranes (MAMs). Biochim Biophys Acta Bioenerg, 2017. 1858(8): p. 615-627.
- Florea, A.-M. and D. Büsselberg, Cisplatin as an Anti-Tumor Drug: Cellular Mechanisms of Activity, Drug Resistance and Induced Side Effects. Cancers, 2011. 3(1): p. 1351-1371.
- Murthy, D., K.S. Attri, and P.K. Singh, Phosphoinositide 3-Kinase Signaling Pathway in Pancreatic Ductal Adenocarcinoma Progression, Pathogenesis, and Therapeutics. Front Physiol, 2018. 9: p. 335.
- Santarpia, L., S.M. Lippman, and A.K. El-Naggar, Targeting the MAPK-RAS-RAF signaling pathway in cancer therapy. Expert Opin Ther Targets, 2012. 16(1): p. 103-19.
- Park, H.K., et al., Induction of apoptosis by laminarin, regulating the insulin-like growth factor-IR signaling pathways in HT-29 human colon cells. Int J Mol Med, 2012. 30(4): p. 734-8.